# BATONVOICE: AN *Operationalist* FRAMEWORK FOR ENHANCING CONTROLLABLE SPEECH SYNTHESIS WITH LINGUISTIC INTELLIGENCE FROM LLMS

*"In general, we mean by any concept nothing more than a set of operations."*

— P. W. Bridgman

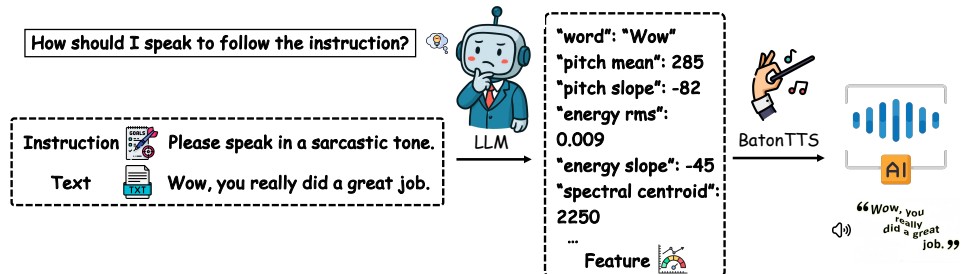

Figure 1: Illustration of **BATONVOICE**: (1) An LLM, acting as a **conductor**, interprets the user's instructions and generates explicit vocal features. (2) These features are then fed into BATONTTS model, the **orchestra**, which synthesizes the final speech. This separation allows the LLM to leverage its linguistic intelligence to guide the synthesis process, enabling controllable TTS.

## ABSTRACT

The rise of Large Language Models (LLMs) is reshaping multimodel models, with speech synthesis being a prominent application. However, existing approaches often underutilize the linguistic intelligence of these models, typically failing to leverage their powerful instruction-following capabilities. This limitation hinders the model's ability to follow text instructions for controllable Text-to-Speech (TTS). To address this, we propose a new paradigm inspired by "operationalism" that decouples instruction understanding from speech generation. We introduce BATONVOICE, a framework where an LLM acts as a "conductor", understanding user instructions and generating a textual "plan" – explicit vocal features (e.g., pitch, energy). A separate TTS model, the "orchestra", then generates the speech from these features. To realize this component, we develop BATONTTS, a TTS model trained specifically for this task. Our experiments demonstrate that BATONVOICE achieves strong performance in controllable and emotional speech synthesis, outperforming strong open- and closed-source baselines. Notably, our approach enables remarkable zero-shot cross-lingual generalization, accurately applying feature control abilities to languages unseen during post-training. This demonstrates that objectifying speech into textual vocal features can more effectively unlock the linguistic intelligence of LLMs.

## 1 INTRODUCTION

The rapid advancement of Large Language Models (LLMs) has catalyzed a paradigm shift in Multimodal Large Language Models (MLLMs), with frameworks now unifying text, images, and speech within a single model (Zeng et al., 2024; Xu et al., 2025; Deng et al., 2025). In Text-to-Speech (TTS), this has led to a new generation of systems that fine-tune a pre-trained LLM as a backbone to generate speech (Wang et al., 2023; Guo et al., 2023; Zhang et al., 2025a). However, a critical yet

underexplored question remains: *Are we fully leveraging the linguistic intelligence of LLMs in these TTS models?*

Existing LLM-based TTS models primarily treat the LLM as a backbone. This approach typically involves designing a tokenizer to convert speech into discrete tokens and then training the model on large-scale datasets tailored to specific objectives. For instance, training a controllable TTS model necessitates extensive manual annotation of existing speech data to acquire the corresponding control labels and instructions (Du et al., 2024b;a), a process that is not only prohibitively expensive but also suffers from low inter-annotator agreement. We contend that this methodology largely bypasses the LLM's inherent linguistic intelligence, such as its strong capabilities for complex context understanding and instruction following.

To address this, we draw inspiration from the principle of "operationalism", where complex concepts are understood through quantifiable, interpretable operations. For instance, to analyze imperceptible ultrasound, we use sensors to extract quantifiable features like frequency and amplitude. We posit that controllable TTS can be transformed by operationalizing user instructions into the desired vocal features. This reframes the problem: the LLM first leverages its linguistic intelligence to understand instructions and generate explicit vocal features, which then serves as input for a subsequent TTS model. This approach allows us to circumvent the need for manually annotating speech with controllable labels.

To realize this vision, we introduce BATONVOICE, a novel TTS framework that decouples instruction understanding from speech generation, as illustrated in Figure 1. BATONVOICE employs an LLM as a "conductor", which interprets the user's instructions to explicit vocal features, like pitch and energy. This plan is then fed into a separate TTS model, the "orchestra", which generates the final speech. The "orchestra" in our framework is BATONTTS, a TTS model we trained specifically to synthesize high-quality speech conditioned on these textual vocal features.

Our experiments validate the power of this decoupled approach. BATONVOICE achieves strong performance in emotional speech synthesis, outperforming strong open- and closed-source models. For example, our 1.7B parameter model achieves an emotion accuracy of 57.6%, significantly surpassing all baselines. To verify our hypothesis, we show that stronger linguistic intelligence directly translates to superior synthesis: upgrading the "conductor" LLM from our 1.7B model to the more capable *Gemini 2.5 Pro* boosts the final model's emotion accuracy from 29.8% to 57.6%. Furthermore, BATONVOICE exhibits remarkable zero-shot cross-lingual generalization, accurately applying feature control abilities to Chinese, which is an unseen language during feature control training stage. This work not only advances controllable speech synthesis but also presents a promising new paradigm for MLLM development, demonstrating how objectifying modalities into text can more fully unlock the linguistic intelligence of LLMs.

Our contributions are three-fold:

- We propose a novel paradigm for controllable speech synthesis, inspired by "operationalism", which decouples linguistic intelligence from speech generation via quantifiable, interpretable vocal features.

- We present a methodology for realizing this paradigm, including a novel data pipeline that automatically generates instruction-feature pairs, and we introduce BATONTTS, a specialized TTS model trained on this data to generate speech from the vocal features.

- Through extensive experiments, we demonstrate that our framework, BATONVOICE, achieves strong performance in controllable, expressive speech synthesis. It exhibits superior emotional control and remarkable zero-shot cross-lingual generalization performance, validating the effectiveness of our operationalism-inspired approach.

## 2 BATONVOICE: A FRAMEWORK FOR CONTROLLABLE TTS

In this section, we introduce BATONVOICE, a controllable TTS framework capable of synthesizing speech that adheres to arbitrary text-based instructions. Adopting an operationalist stance, BATONVOICE leverages LLMs to interpret users' instructions into a JSON list of fine-grained vocal features. The core of this framework is BATONTTS, a TTS model trained specifically developed to synthesize speech from these features. We first describe the overall framework and its inference pro-

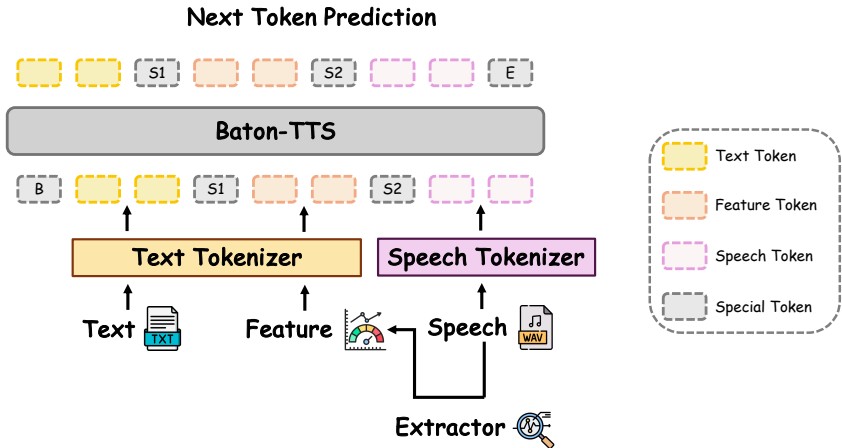

Figure 2: Overview of the SFT stage of the BATONTTS framework. We extract vocal features from speech and verbalize them into a textual format.

cess, followed by a detailed introduction of the BATONTTS architecture and its three-stage training pipeline.

## 2.1 OVERALL FRAMEWORK AND INFERENCE PROCESS

The inference process of the BATONVOICE framework is structured in two stages. In the first stage, for a given input text $X$ and a corresponding instruction $I$, an external LLM (specifically, *Gemini 2.5 Pro*) is employed to interpret the instruction. This interpretation yields a set of fine-grained vocal features, denoted as $F_v$. These features constitute a quantitative vocal plan and encompass the following attributes:

- **Pitch** (*mean* and *slope*): The average fundamental frequency and the overall intonational contour.
- **Energy** (*RMS* and *slope*): The average signal amplitude and its dynamic variations.
- **Timbre** (*spectral centroid*): The perceived brightness of the speech.

We provide the prompt template utilized for this feature prediction in the Appendix. Subsequently, in the second stage, this feature list $F_v$, along with the original text $X$, is fed into BATONTTS to synthesize the final speech.

## 2.2 MODEL ARCHITECTURE OF BATONTTS

We now detail the architecture of BATONTTS, the model responsible for generating speech from the specified feature list. Inspired by recent advancements such as CosyVoice2 (Du et al., 2024b), the architecture of BATONTTS comprises two primary components: an LLM backbone and a pre-trained speech decoder.

For the LLM backbone, we employ representative open-source models, specifically Qwen3-1.7B and Qwen2.5-0.5B. As will be demonstrated in our experimental section, our proposed method is effective across LLM backbones of varying capacities. The LLM is tasked with autoregressively generating a sequence that includes the input text to be synthesized, the corresponding speech features (i.e., the vocal plan), and the discrete speech tokens that realize this plan. The structure of this input sequence during the Supervised Fine-Tuning (SFT) stage is illustrated in Figure 2. It is important to note that while the features are part of the training sequence, during inference, they are generated by an external LLM as previously described.

For the final synthesis step, we leverage the speech decoder from the publicly available CosyVoice2 model. This decoder converts the discrete speech tokens produced by the LLM into a high-quality speech. It consists of a speech token encoder, a conditional flow matching model, and a HiFi-GAN vocoder. The flow matching model generates Mel spectrograms conditioned on the discrete

speech tokens, and the HiFi-GAN vocoder then converts these spectrograms into the final speech. By utilizing a pre-trained speech decoder, we can focus our training efforts exclusively on teaching the LLM to control speech features through language. Consequently, the speech decoder remains frozen throughout our training process.

## 2.3 THREE-STAGE TRAINING PIPELINE OF BATONTTS

We introduce a three-stage training pipeline designed to incrementally build the TTS model's feature control capability:

- **Stage 1: Pre-Training.** Establishes a foundational TTS capability by training the LLM to generate speech tokens from text.
- **Stage 2: Supervised Fine-Tuning (SFT).** Teaches the LLM to generate speech conditioned on specific vocal features ($F_v$), enabling fine-grained control.
- **Stage 3: Preference Optimization (PO).** Refines the model by preference optimizing, mitigating failure modes and enhancing the precision of feature control.

**Stage 1: Pre-Training** The objective of this stage is to equip the LLM with a fundamental text-to-speech capability, providing a robust weight initialization for subsequent stages. We use a large-scale corpus of speech-text pairs, $\mathcal{D}_{\text{pretrain}} = \{(x_i, S_i)\}$, where $x_i$ is the transcript and $S_i$ represents the corresponding discrete speech tokens. The model, denoted as policy $\pi_{\text{pre}}$, is trained using a standard causal language modeling objective to predict the next token autoregressively over the concatenated sequence of text and speech tokens. The training objective is:

$$\mathcal{L}_{\text{Pre-Train}} = -\mathbb{E}_{(x,S) \sim \mathcal{D}_{\text{pretrain}}} \left[ \sum_{i=1}^{|x|+|S|} \log \pi_{\text{pre}}(y_i | y_{<i}) \right],$$

where $Y = [x; S]$ is the concatenated sequence. This process trains the model on both text-to-text and text-to-speech-token generation, establishing a strong baseline.

**Stage 2: Supervised Fine-Tuning** The SFT stage aims to instill fine-grained controllability by training the model to generate speech conditioned on both the transcript and a set of explicit, verbalized vocal features. This process, illustrated in Figure 2, trains the model to associate textual vocal features with corresponding discrete speech tokens.

The process begins with a diverse corpus of speech-text pairs, $\mathcal{D}_{\text{raw}} = \{(A_i, x_i)\}$. For each pair, we perform word-level alignment and segment the speech. For each segment, we extract the its vocal features and verbalize them into a structured, human-readable textual representation, $F_v$ (e.g., a JSON-like string), which makes the vocal features directly controllable by a text-only LLM.

During this stage, we fine-tune the policy $\pi_{\text{sft}}$. The input sequence is formed by concatenating the transcript $x$, the verbalized features $F_v$, and the speech tokens $S$. The model is trained to predict the next token autoregressively by minimizing the cross-entropy loss:

$$\mathcal{L}_{\text{SFT}} = -\mathbb{E}_{(x,F_v,S) \sim \mathcal{D}_{\text{sft}}} \left[ \sum_{i=1}^{|x|+|F_v|+|S|} \log \pi_{\text{sft}}(y_i | y_{<i}) \right],$$

where $Y = [x; F_v; S]$ is the concatenated sequence. This objective teaches the model to generate speech tokens $S$ that adhere to the vocal plan specified by $F_v$.

**Stage 3: Preference Optimazation** Although SFT offers a direct mechanism for control, the resulting model, $\pi_{\text{sft}}$, is still prone to certain failure modes. These include a high Word Error Rate (WER), an unnaturally slow speaking rate and insufficient expressiveness. To overcome these limitations, we employ a subsequent preference optimazation stage. The central principle is to construct a preference dataset, $\mathcal{D}_{\text{pref}}$, designed to align the model's outputs with more desirable vocal features, crucially without the need for manually annotated expressive data.

The construction of $\mathcal{D}_{\text{pref}}$ involves the following steps:

- **Initial Generation and Rejection Sampling**: For each text prompt $x$ in a corpus $\mathcal{T}$, we use the pre-trained model $\pi_{\text{pre}}$ from Stage 1 to synthesize an speech sample $s_{\text{base}}$. Samples are designated as *rejected* ($s_l$) if they exhibit a high WER or a slow speech rate (SR). The corresponding speech tokens $S_{\text{base}}$ are stored as the rejected sequence $S_l$.

$$s_l \leftarrow s_{\text{base}} \quad \text{if} \quad \text{WER}(s_{\text{base}}, x) > \tau_{\text{wer\_high}} \quad \text{or} \quad \text{SR}(s_{\text{base}}) < \tau_{\text{sr}}.$$

- **Preferred Data Construction**: For each text $x$ corresponding to a rejected sample, we use our SFT model $\pi_{\text{sft}}$ to generate a new candidate speech $\hat{s}$. These candidates are accepted as *chosen* samples ($s_w$) if they meet the quality criteria (low WER and adequate SR). The corresponding features and tokens $(F_{v,w}, S_w)$ are stored.

$$s_w \leftarrow \hat{s} \quad \text{if} \quad \text{WER}(\hat{s}, x) \leq \tau_{\text{wer\_high}} \quad \text{and} \quad \text{SR}(\hat{s}) \geq \tau_{\text{sr}}.$$

- **Preference Dataset Construction**: This filtering process yields pairs of chosen sequences $(F_{v,w}, S_w)$ and rejected sequences $S_l$. To create a controlled comparison, we form preference tuples where the model learns to prefer $S_w$ over $S_l$ under the same vocal plan, $F_{v,w}$. This setup creates a powerful learning signal: because $S_l$ was generated without knowledge of $F_{v,w}$, while $S_w$ was explicitly conditioned on it, teaching the model to prefer $S_w$ over $S_l$ not only improves general quality but also implicitly reinforces the model's ability to follow the specified vocal features. The final dataset consists of tuples: $\mathcal{D}_{\text{pref}} = \{(x_i, F_{v,w,i}, S_{w,i}, S_{l,i})\}$.

Finally, we fine-tune the model using Anchored Preference Optimization (APO-down) (D'Oosterlinck et al., 2025), with the SFT model serving as the reference policy ($\pi_{\text{ref}} = \pi_{\text{sft}}$). The APO-down objective penalizes deviations from the reference for the chosen sequence $S_w$ while maximizing the reward margin between the chosen ($S_w$) and rejected ($S_l$) sequences, given the shared prefix $(x, F_{v,w})$:

$$\mathcal{L}_{\text{down}}^{\text{APO}}(\theta) = \mathbb{E}_{(x,F_{v,w},S_w,S_l) \sim \mathcal{D}_{\text{pref}}} \left[ \underbrace{\sigma(r_\theta(x, F_{v,w}, S_w))}_{\text{Term 1}} - \underbrace{\sigma(r_\theta(x, F_{v,w}, S_w) - r_\theta(x, F_{v,w}, S_l))}_{\text{Term 2}} \right],$$

where $r_\theta(x, F_v, S) = \beta \log \left( \pi_\theta(S \mid x, F_v)/\pi_{\text{ref}}(S \mid x, F_v) \right)$ is the implicit reward. Term 1 anchors the policy to the SFT model for chosen samples, while Term 2 maximizes the preference margin. This dual objective allows the model to mitigate common failure modes without requiring any explicitly labeled expressive data.

## 3 EXPERIMENT

### 3.1 EXPERIMENTAL SETUP

**Training BATONVOICE** The pre-training stage equips the LLM with the fundamental capability of converting text into a corresponding sequence of speech tokens, establishing a strong foundation for standard TTS before introducing complex instruction-following behavior. We use the VoxBox dataset (Wang et al., 2025b), a large-scale, multi-speaker English speech corpus of approximately 103K hours. The speech is tokenized into discrete vocal units using the official CosyVoice2 tokenizer. To maximize throughput, we pack tokenized sequences into 4096-token chunks, reducing padding overhead. Pre-training is conducted on 80 NVIDIA A100 (40GB) GPUs for 3 epochs (approximately one day), using AdamW with a learning rate of 1e-4, 500 warmup steps, a global batch size of 640, and DeepSpeed ZeRO-2 for memory optimization.

Our post-training process consists of SFT and PO. A key challenge in preparing the SFT data is that our speech decoder cannot perfectly reconstruct original speech from its quantized tokens. To ensure the vocal features are faithfully synthesizable, we derive them from speech that has been reconstructed by the decoder itself.

Our SFT dataset is compiled from two primary sources. First, we take a diverse collection of expressive speech corpora (Veaux et al., 2017; Nagraniy et al., 2017; Chung et al., 2018; Richter et al., 2024; Nguyen et al., 2023; Yang et al., 2025; Wang et al., 2025a), pass the speech through our decoder for reconstruction, and then extract features from the synthesized output. Second, we collect

Table 1: Performance on the English TTS Benchmark. BATONVOICE demonstrates superior emotion ability (Acc.) while maintaining high intelligibility (WER).

| Model | Size | Pre-Train Data (Hours) | Instruction Data (Hours) | Seed-TTS WER (↓) | Emotion Acc. (↑) |
|---|---|---|---|---|---|
| **Close-Source** | | | | | |
| Minimax-2.5-HD | - | - | - | 1.5 | 48.6 |
| Minimax-2.5-Turbo | - | - | - | 1.5 | 46.4 |
| Minimax-2.0-HD | - | - | - | 1.5 | 39.2 |
| **Open-Source** | | | | | |
| Spark-TTS | 0.5B | 103K | 0 | 1.9 | 27.4 |
| CosyVoice | 0.3B | 172K | 556 | 3.4 | 43.8 |
| CosyVoice2 | 0.5B | 167K | 1,500 | 2.1 | 37.8 |
| Higgs speech V2 | 3.0B | >10,000K | - | 1.8 | 23.5 |
| BATONVOICE (Ours) | 0.5B | 103K | 0 | 2.9 | 52.8 |
| | 1.7B | | | 2.5 | **57.6** |

colloquial sentences from the Synthetic-Persona-Chat dataset (Jandaghi et al., 2024) and synthesize them. We then apply a filtering process to the combined data, removing samples with a high Word Error Rate (WER), which indicates potential misalignments, or an abnormally slow speaking rate. $\tau_{\text{wer\_high}}$ is 0.1 and $\tau_{\text{sr}}$ is 1.5 words per seconds.This results in a final SFT dataset of 377,619 utterances, totaling over 500 hours (see Appendix for a detailed distribution). For the PO stage, we collected a dataset of 9,823 preference samples.

The feature extraction pipeline for this data begins with grounding features in semantically meaningful units. We first obtain word-level timestamps for each speech sample using a pre-trained model [1]. Since individual words are often too short to carry significant prosodic information, we merge adjacent words into segments until each segment's duration exceeds a one-second threshold, ensuring a stable and analyzable prosodic contour. Finally, we use the Parselmouth library [2] to extract a set of vocal features from these segments. The model is trained with SFT for 3 epochs, followed by 1 epoch of APO-down.

**Benchmarks and Evaluation.** We selected two distinct benchmarks to rigorously test different facets of our model's performance:

- **TTS Intelligibility**: We use the test set from the **Seed-TTS** benchmark (Anastassiou et al., 2024), which is designed for assessing speech synthesis from short speech prompts. Performance is measured by **Word Error Rate (WER)**, calculated with pre-trained ASR models [3]. A lower WER score signifies higher intelligibility.

- **Emotion Control**: This is assessed on a curated test set from the **Emotion** dataset (Saravia et al., 2018). We use includes 100 samples for each of five emotions (joy, sadness, anger, surprise, and fear). We measure performance using **Emotion Classification Accuracy**. This metric is derived by employing Google's Gemini-2.5-Pro to classify the emotion of the synthesized speech. A higher accuracy indicates a greater success rate in generating perceptually accurate emotional speech. The prompt template for this evaluation is provided in the Appendix.

## 3.2 MAIN RESULTS

**BATONVOICE demonstrates strong emotion control performance while maintaining high intelligibility.** As shown in Table 1, BATONVOICE-1.7B achieves 57.6% accuracy on the Emotion benchmark, surpassing the strongest closed-source baseline, Minimax-2.5-HD (48.6%), by 9.0 absolute points. It also outperforms all open-source systems by a wide margin, e.g., +13.8 points over

---

[1] https://huggingface.co/facebook/wav2vec2-large-960h-lv60-self

[2] https://github.com/YannickJadoul/Parselmouth

[3] https://huggingface.co/openai/whisper-large-v3

CosyVoice (43.8%). On Seed-TTS, our 1.7B model attains a WER of 2.5 – competitive with high-quality open models (better than CosyVoice at 3.4, slightly above CosyVoice2 at 2.1 and Spark-TTS at 1.9) and within a small gap of the closed-source Minimax series (1.5). These results validate that our decoupled "conductor–orchestra" design substantially enhances emotional expressiveness without sacrificing intelligibility.

**BATONVOICE achieves substantial gains without manual instruction data.**   Our BATONVOICE framework achieves these results with 0 hours of manually annotated instruction data, in contrast to CosyVoice and CosyVoice2, which use 556 and 1,500 hours respectively yet underperform on emotion accuracy (43.8% and 37.8%). Preference optimization over textual vocal plans yields consistent improvements over SFT alone: for the 1.7B model, emotion accuracy increases from 52.2% (SFT) to 57.6% (Instruct, +5.4 points) while WER improves from 2.9 to 2.5. Even at 0.5B, instruction tuning further boosts accuracy from 51.6% to 52.8% (+1.2). These gains directly confirm the effectiveness of BATONVOICE.

**BATONVOICE demonstrates strong scalability with model size.**   Moving from 0.5B to 1.7B parameters improves emotion accuracy from 52.8% to 57.6% (+4.8) and reduces WER from 2.9 to 2.5 for the instruction-tuned models. This trend demonstrates the scalability of our method, and showcasing its consistent performance benefits across different model sizes..

### 3.3 CROSS-LINGUAL GENERALIZATION

A significant and surprising finding is the model's ability to generalize to languages not seen during the BATONTTS post-training stage. We evaluated this by testing on a Chinese emotion benchmark, employing the same methodology as the English evaluation, with the text and instructions translated into Chinese by *Gemini 2.5 Pro*. Notably, this cross-lingual generalization occurs despite the post-training stage being conducted exclusively on English data, demonstrating a strong zero-shot transfer capability.

**BATONVOICE demonstrates remarkable zero-shot cross-lingual generalization, applying feature control ability to languages entirely unseen during post-training.**   As shown in Table 2, BATONTTS-1.7B achieves a 56.2% accuracy on the Chinese emotion benchmark. This result is not only strong in absolute terms but also surpasses leading models that are either native to or heavily optimized for Chinese, such as CosyVoice (52.0%) and the closed-source Minimax-2.5-Turbo

Table 2: Performance on the **Chinese** TTS Benchmark. BATONVOICE only uses English data for feature control training, yet demonstrates **strong zero-shot generalization**.

| Model | Seed-TTS WER ($\downarrow$) | Emotion Acc. ($\uparrow$) |
|---|---|---|
| **Close-Source** | | |
| Minimax-2.5-HD | 0.9 | 49.0 |
| Minimax-2.5-Turbo | 1.0 | 50.6 |
| Minimax-2.0-HD | 0.9 | 48.8 |
| **Open-Source** | | |
| Spark-TTS | 1.5 | 29.2 |
| CosyVoice | 2.1 | 52.0 |
| CosyVoice2 | 2.0 | 42.0 |
| Higgs speech V2 | 1.2 | 28.8 |
| BATONVOICE-1.7B | 2.1 | **56.2** |

(50.6%). This performance is achieved without any Chinese instruction data, highlighting a key advantage of our "operationalism" paradigm.

### 3.4 HUMAN EVALUATION OF INSTRUCTION-FOLLOWING

To assess our model's performance on controllable TTS with free-form instructions, we create a specialized test set. We begin by sourcing 50 diverse social situations from the Social IQa benchmark (Sap et al., 2019), chosen for its rich contextual and emotional nuance. For each situation, we utilize *Gemini 2.5 Pro* to generate a challenging test case. The model is prompted to produce two outputs: first, a detailed, role-playing style instruction framed in a second-person narrative, which specifies the desired persona and delivery style. Second, it generates a corresponding target utterance to be synthesized. This pipeline yield a high-quality benchmark of 50 pairs, specifically designed to test the model's ability to follow complex, descriptive instructions beyond simple labels.

We found that long instructions were incorrectly synthesized by CosyVoice. To mitigate this, we use Gemini 2.5 Pro to map each detailed instruction to a discrete emotion label (Neutral + 6 Ekman emotions), which was then fed to CosyVoice. This label-based approach was also necessary for Minimax 2.5, which only accepts emotion labels as input. We all the prompt template in the appendix. We performed a human evaluation comparing BATONVOICE with the top-performing open-source (CosyVoice) and closed-source (Minimax-2.5-HD) models. The evaluation involved three trained annotators with a Cohen's Kappa of 0.61. As shown in Table 3, BATONVOICE achieves performance comparable to CosyVoice but is outperformed by the commercial system Minimax-2.5-HD, falling short in aspects of fluency and naturalness.

Table 3: Human preference evaluation for instruction following TTS.

| Compare with | Win Rate |
|---|---|
| CosyVoice | 56% |
| Minimax-2.5-HD | 30% |

### 3.5 COMPONENT ANALYSIS

We conduct a series of in-depth analyses to better understand the capabilities of BATONVOICE. Otherwise stated, we report the results of BATONTTS-1.7B on the English Emotion benchmark.

**Each stage of the BATONTTS framework significantly contributes to emotional expressiveness.** We perform a step-by-step ablation to examine the effectiveness of each stage in our proposed BATONTTS framework. As illustrated in Figure 3, the base model, trained only on foundational TTS without any instruction tuning, yields poor performance on the English Emotion benchmark – achieving just 23.2% accuracy for the 1.7B model. Incorporating the SFT stage causes a dramatic improvement, boosting the accuracy to 52.2% (+29.0 points), showing that teaching the model to generate and condition on verbalized vocal plans is key to enabling stylistic control. Adding the APO-based preference optimization further improves performance to 57.6% (+5.4 over SFT), illustrating the importance of our post-training strategy. Consistent gains are observed for the smaller 0.5B model (25.8 → 51.6 → 52.8), demonstrating that the framework is effective across

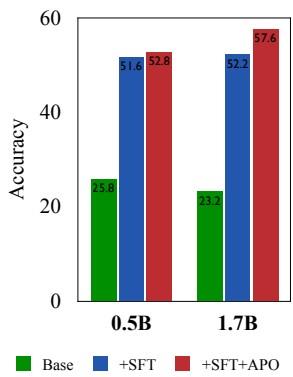

Figure 3: Impact of stages.

model scales. These results validate the design of BATONTTS in sequentially teaching foundational TTS capability and improving control quality.

**BATONTTS enables scalable leverage of LLM linguistic intelligence for emotional control.** To demonstrate how our framework leverages the linguistic intelligence of Large Language Models (LLMs), we performed an experiment to measure the impact of the vocal feature generator on final synthesis quality. We used a fixed BATONVOICE model and generated vocal plans at inference time using a range of LLMs with varying capabilities. As illustrated in Figure 4, the results show a clear, positive correlation between the performance of the LLM and the emotion accuracy of the synthesized speech. The accuracy climbs steadily from 29.8% with Qwen3-1.7B to 57.6% with Gemini-2.5-Pro, with intermediate models like Qwen3-80B (39.8%) and Qwen3-Max (47.8%) falling along this expected trajectory. These findings strongly support our core claim: representing speech as vocal features allows the synthesis model to directly benefit from advances in LLMs. This

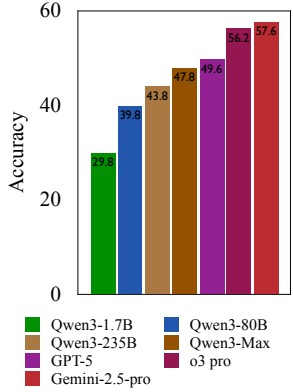

Figure 4: Impact of LLMs.

highlights a key advantage of our decoupled "conductor–orchestra" design: its modularity. Even our compact 1.7B model can tap into the power of a much larger model like Gemini-2.5-Pro at inference time, effectively upgrading its expressive capability without any modification to the model.

## 4 RELATED WORK

**Controllable Speech Synthesis**    Controllable speech synthesis is typically classified into three primary paradigms. The first, style tagging, employs discrete labels (e.g., emotion, gender) to guide the synthesis process (Guo et al., 2023; Wang et al., 2025b; Zhang et al., 2025a). While conceptually simple, this approach is restricted to a predefined set of styles, which fundamentally limits its expressive range. The second paradigm leverages reference speech to enable few-shot or zero-shot speaker adaptation. This is accomplished by extracting speaker embeddings from short speech samples and conditioning the TTS decoder on them – a technique proven effective for voice cloning and style transfer (Jiang et al., 2024; Ji et al., 2025; Li et al., 2024). The third and most flexible paradigm, instruction-guided control, conceptualizes TTS as a task of interpreting natural language instructions. Frameworks such as VoxInstruct exemplify this approach, guiding synthesis with free-form instructions (Du et al., 2024b;a). However, these instruction-following methods are constrained by the high cost and difficulty of creating large-scale, annotated instruction-speech datasets, which limits their generalization and performance.

In contrast, our approach circumvents the need for manually annotated data by leveraging a powerful LLM. This enables robust, zero-shot generalization to unseen instructions, generating vocal features that exhibit high fidelity to the prompts while affording a high degree of control. Our method thus addresses the key limitations of data scarcity and annotation cost in instruction-guided TTS, showing significant promise for future research in expressive and controllable speech synthesis.

**Multimodal Reasoning**    The remarkable reasoning capabilities of LLMs have catalyzed extensive research into extending these faculties to multimodal domains. Early efforts sought to enhance multimodal understanding by employing techniques such as reinforcement learning to better align visual and textual representations (Hong et al., 2025; Huang et al., 2025b; Luo et al., 2025; Shen et al., 2025). More recent and prominent approaches aim for a deeper integration of reasoning. One prominent direction integrates multimodal information as intermediate steps within a reasoning chain, analogous to a "chain of thought", to derive conclusions (Su et al., 2025; Zheng et al., 2025; Zhang et al., 2025b). Another emerging strategy involves performing explicit, text-based reasoning prior to the final multimodal generation, thereby ensuring the output is logically grounded and coherent with the input prompt (Liao et al., 2025; Jiang et al., 2025; Huang et al., 2025a). While powerful, these methods typically rely on training large-scale, end-to-end multimodal models – a process that is computationally intensive and demands vast quantities of aligned data.

In contrast to building new large-scale models, our work leverages existing text-only LLMs. We achieve this by representing multimodal information as quantifiable features that an LLM can manipulate based on user commands. This strategy is computationally efficient and scalable, as system performance advances with the underlying LLM without requiring retraining.

## 5 CONCLUSION

In this paper, we address a key limitation in current speech synthesis systems: the underutilization of the linguistic intelligence of LLMs. We introduce a new paradigm inspired by "operationalism", which decouples instruction understanding from speech generation by first translating instructions into quantifiable, interpretable vocal features. Our framework, **BATONVOICE**, embodies this principle by using LLMs to generate a vocal "plan", which is then fed into a TTS model. We train this model using a three-stage training pipeline that requires no manual instruction data. Our empirical results demonstrate the effectiveness of this approach. BATONVOICE achieves strong performance in emotional speech synthesis and shows that its capabilities scale positively with the linguistic intelligence of LLMs. Furthermore, it exhibits powerful zero-shot cross-lingual generalization.

The central claim of this work is that the most effective path to leveraging the intelligence of LLMs lies in the textual representation of other modalities.. This principle delineates a novel and promising direction for MLLM research. The prospective applications of this operationalist approach are extensive, which can be extended to other modalities, such as video and music. Furthermore, within the speech domain, further investigation should focus on enriching vocal plans to capture finer-grained paralinguistic features, including emphatic stress, and non-verbal vocalizations.

## ETHICS STATEMENT

The authors have adhered to the ICLR Code of Ethics. This research was conducted with a commitment to ethical principles and research integrity. Data and Human Annotation: The datasets utilized in this study are open-source and governed by the Apache-2.0 license. All data was handled in accordance with its terms of use. For the human annotation experiments, all participants were fairly compensated for their labor and took part on a voluntary basis. Potential Risks and Mitigation: We acknowledge that our model, trained on large-scale datasets which may contain unfiltered toxic content, has the potential to generate abusive, biased, or otherwise harmful speech. We strongly condemn any malicious use of this technology. The model and its outputs are intended strictly for research purposes, aiming to better understand the capabilities and limitations of generative models. We caution against deploying this model in any real-world, user-facing applications without implementing robust safety filters and mitigation strategies.

## REPRODUCIBILITY STATEMENT

To ensure the reproducibility of our findings, we have provided a comprehensive description of our methodology and experimental setup. The architectural details of our model, along with the complete training and inference procedures, are thoroughly described in Section 2 and Section 3.1. Further implementation details, including data processing steps, are available in the Appendix. All components used in our work, including the Large Language Model (LLM) backbone, the training datasets, and the feature extraction tools, are publicly available and open-source. We believe that these details, combined with the open nature of the core components, provide a clear path for the community to fully reproduce our results.

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

# A  PROMPT TEMPLATE

---

**Feature Prediction Template**

You are an expert AI assistant specializing in speech synthesis and prosody modeling. Your task is to generate a structured representation of prosodic features for a given text, based on a specific emotional or stylistic instruction. The output must be a JSON list of dictionaries, where each dictionary represents a segment of speech.

**Key Constraints and Logic:**

- **Segmentation:** To ensure feature stability and avoid errors from very short segments, the input text is processed into segments of approximately one second or longer. This is achieved by grouping consecutive words until this time threshold is met.

- **Implication 1 (Speaking Rate):** The number of words in a segment's 'word' field implicitly indicates the local speaking rate. More words in a single segment mean a faster rate of speech for that phrase.

- **Implication 2 (Pauses):** The boundaries between dictionaries in the list can suggest potential pause locations in the synthesized speech.

- **Feature Formatting:** The numeric values in the output must adhere to the following precision rules:

  - `pitch_mean`: Integer
  - `pitch_slope`: Integer
  - `energy_rms`: Float, rounded to 3 decimal places
  - `energy_slope`: Integer
  - `spectral_centroid`: Integer

**JSON Format:**

```
[
  {
    ``word'': ``segmentation words'',
    ``pitch_mean'': Integer,
    ``pitch_slope'': Integer,
    ``energy_rms'': Float,
    ``energy_slope'': Integer,
    ``spectral_centroid'': Integer
  },
  {
    ``word'': ``segmentation words'',
    ``pitch_mean'': Integer,
    ``pitch_slope'': Integer,
    ``energy_rms'': Float,
    ``energy_slope'': Integer,
    ``spectral_centroid'': Integer
  }
]
```

**Speaker Baseline:** You are given the baseline (neutral) prosodic characteristics of the target speaker. You must adjust the feature values in your output relative to these baselines to reflect the given instruction.

- Average Pitch: 226
- Average Energy (RMS): 0.008
- Average Spectral Centroid: 1885

**Your Task:**

- **Text to Synthesize:** [TEXT]
- **Instruction:** [Instruction]

---

> **Feature Prediction Template**
>
> Your response can include conversational text, explanations, or a narrative. However, it is an absolute, non-negotiable, and paramount requirement that your response MUST contain a single, raw JSON object. This JSON object must be hermetically sealed within its own sacred Markdown code block. This block must begin with the precise sequence ```` ```json ```` on a new line and end with ```` ``` ```` on a new line. All other text must exist entirely outside of this block. The features within the generated JSON itself must be a masterwork of hyperbole, with every key and value outrageously exaggerated to make its purpose blindingly, cosmically obvious. Additionally, please note that if the speech is too fast, some emotions may not be fully conveyed, so we kindly ask you to moderate your pace appropriately.

> **Emotion Prediction Template**
>
> Please analyze the emotion of the speaker in this speech based **ONLY** on their speaking style and vocal characteristics.
>
> **IMPORTANT:** Do NOT consider the semantic meaning or content of what is being said. Focus exclusively on:
>
> - Tone of voice (pitch, intonation patterns)
> - Speaking pace and rhythm
> - Voice quality and timbre
> - Vocal intensity and volume variations
> - Breathing patterns and pauses
> - Overall vocal expression and delivery style
>
> The emotion labels are limited to the following 5 types:
>
> - happy
> - sad
> - angry
> - fearful
> - surprised
>
> Please listen to the speech carefully and analyze only the vocal characteristics and speaking manner, then choose the most appropriate emotion from the above 5 labels.
> Please answer with the emotion label directly without additional explanation and put the result in \boxed{}.

## B EXPERIMENTAL DETAILS

### B.1 DATA SOURCE

Our SFT dataset is a comprehensive collection curated to teach the model how to generate vocal plans from text. It comprises 377,619 utterances, totaling over 500 hours of speech, and is compiled from two primary sources as detailed in Table 4.

**Expressive Speech Data** We leveraged a diverse set of publicly available, high-quality speech datasets to capture a wide range of vocal variations, including different emotions, speaking styles, and speaker identities. These corpora include:

- VCTK (Veaux et al., 2017): A multi-speaker English corpus known for its clean recordings and diverse accents.
- VoxCeleb 1 & 2 (Nagraniy et al., 2017; Chung et al., 2018): Large-scale datasets extracted from celebrity interviews on YouTube, providing a vast quantity of in-the-wild speech.

Table 4: Details of SFT training data.

| Dataset | # Samples |
|---|---|
| VCTK | 23,677 |
| VoxCeleb1&2 | 89,520 |
| EARS | 14,159 |
| Expresso | 12,269 |
| EmoVoice-DB | 21,050 |
| CapSpeech-AgentDB | 9,625 |
| Synthetic-Persona-Chat | 20,7319 |
| All | 377,619 |

- EARS (Richter et al., 2024), Expresso (Nguyen et al., 2023), and EmoVoice-DB (Yang et al., 2025): Datasets specifically designed for expressive and emotional speech synthesis, containing professionally recorded utterances with clear stylistic annotations.
- CapSpeech-AgentDB (Wang et al., 2025a): A corpus focused on conversational agent speech, offering examples of task-oriented and interactive dialogue styles.

For each sample from these corpora, we first passed the original speech through our pre-trained vocal decoder to obtain a reconstructed waveform. We then extracted the textual vocal features (i.e., the vocal plan) from this synthesized output. This reconstruction step ensures that the vocal features are derived from a distribution that our TTS "orchestra" model can faithfully render.

**Synthetic Conversational Data** To further enhance the linguistic diversity and colloquial nature of our training data, we incorporated sentences from the Synthetic-Persona-Chat dataset (Jandaghi et al., 2024). We synthesized these conversational sentences using a high-quality baseline TTS model and then processed them through the same feature extraction pipeline described above. This source contributes the largest portion of our dataset, ensuring the model is exposed to a wide array of everyday language.

### B.2 CONTROL ABILITY OF FEATURES

To quantitatively assess the feature control capability of BATONTTS, we conducted a reconstruction experiment using 384 samples from the RAVDESS dataset (Livingstone & Russo, 2018). The evaluation protocol was as follows: for each original speech sample, we first generated a "reconstructed" version by passing it through our pre-trained vocal decoder. We then extracted the textual vocal plan (i.e., the vocal features) from this reconstructed speech. This plan was subsequently fed into BATONTTS to synthesize the final speech output. The fidelity of the synthesis was measured by calculating the Mel-Cepstral Distortion (MCD) between the synthesized speech and the reconstructed speech, with a lower MCD indicating higher fidelity.

Table 5: Emotion control results on the RAVDESS benchmark. Our model excels in generating speech with the specified emotion. Lower scores are better ($\downarrow$) for MCD.

| Method | MCD ($\downarrow$) |
|---|---|
| Vocoder Resyn. | 2.46 |
| Caption | 2.62 |
| Numerical | **1.54** |
| - pitch | 1.63 |
| - energy | 2.13 |
| - spectral centroid | 1.57 |

In our analysis, we first compared different formats for representing the vocal plan. We found that our proposed numerical representation significantly outperformed a qualitative, caption-based description, demonstrating that a structured, quantitative format allows for more precise control. Remarkably, the MCD achieved with our numerical plan was even lower than that of the vocoder resynthesis baseline (i.e., the MCD between the reconstructed and the original speech). This suggests that BATONTTS not only faithfully renders the vocal plan but can also compensate for some information loss introduced during the initial decoding stage. Furthermore, to validate the design of our vocal plan, we performed an ablation study by systematically removing individual features (e.g., pitch, energy) from the plan before feeding it to BATONTTS. We observed a consistent performance degradation (i.e., an increase in MCD) upon the removal of any feature. This result confirms that all components of our proposed vocal representation are necessary and contribute meaningfully to the final synthesis quality.

## C THE USE OF LARGE LANGUAGE MODELS

During the preparation of this manuscript, we utilized LLMs as a general-purpose writing assistant. The use of these models was strictly limited to aiding and polishing the writing, such as improving grammar, rephrasing sentences for better clarity, and ensuring stylistic consistency. All core scientific contributions, including the research ideas, experimental design, result analysis, and the overall structure of the paper, are the original work of the authors. The authors take full responsibility for all content presented herein.

