# OpenReview forum: "BatonVoice: An Operationalist Framework for Enhancing Controllable Speech Synthesis with Linguistic Intelligence from LLMs"
_ICLR.cc/2026/Conference — ICLR 2026 Conference Withdrawn Submission_

### Official Review · Reviewer_uXdn · 2025-10-30

**Soundness:** 2
**Presentation:** 3
**Contribution:** 2
**Rating:** 4
**Confidence:** 3

**Summary:**

The paper introduces BatonVoice, a new paradigm for controllable text-to-speech that explicitly leverages the linguistic intelligence of large language models. THe  inspiration comes from operationalism, where the authors propose to decouple instruction understanding from speech generation. In the proposed method, an LLM interprets text-based instructions and outputs explicit vocal features such as pitfch, energy, and spectral centroid in a structure textual form. Then a TTS model synthesizes the final waveform conditioned on these features.

The proposed method indicate that the model could get superior emotional control while maintaining competitive intelligibility. Notably, the system shows zero-shot cross-lingual generalization from English to Chinese and improved controllability without any manual annotated instruction data.

**Strengths:**

- The paper presents a new conceptual framing for controllable TTS, using operationalism as a guiding philosophy to bridge linguistic reasoning and acoustic generation.
- The “conductor–orchestra” design is with good modular design: stronger LLMs (e.g., Gemini 2.5 Pro) immediately improve synthesis quality without retraining.
- The cross-lingual generalization results are particularly impressive and theoretically grounded: since vocal features are language-agnostic, the control mechanism naturally transfers.

**Weaknesses:**

- While emotion accuracy is a reasonable proxy for controllability, it does not cover the breadth of expressive control (e.g., prosodic variation, speaking style, or character-level consistency).
- Human evaluation (Table 3) shows only comparable performance to CosyVoice and worse than Minimax in naturalness, suggesting that emotional control may trade off fluency.
- The philosophical term “operationalism” adds rhetorical appeal, but its operationalization is relatively shallow, essentially describing feature-based decomposition. More justification may be necessary.
- The system heavily relies on pretrained decoders (CosyVoice2) and external LLMs (Gemini 2.5 Pro). It’s unclear how much of the observed performance comes from the proposed framework versus the inherent strength of these components.
- The human evaluation is limited (50 cases, 3 annotators), which may not suffice for strong claims about general controllability.
- The paper briefly mentions model failures (e.g., slow rate, high WER) but lacks systematic qualitative examples or analysis of what types of instructions fail (e.g., nuanced tone, multi-emotion blends).

**Questions:**

- How sensitive is batonvoice to the specific choice and granularity of vocal features (e.g., pitch, energy, spectral centroid)? Would a higher-resolution (frame-level) or lower-resolution (utterance-level) representation alter controllability?
- Could the authors clarify how different combinations of features (e.g., removing pitch or timbre) affect perceived emotion and control quality beyond MCD scores? Are there perceptual trends that match acoustic degradations?
- How does the model ensure alignment between free-form text instructions and the numerical features? Is there any constraint or regularization ensuring that the LLM’s predicted features correspond semantically to the intended expression?
- The human study covers 50 instruction–utterance pairs. Have the authors considered a larger-scale or listener study (e.g., MOS for naturalness and instruction faithfulness)? How consistent were annotators’ preferences across emotions or speakers?
- What types of instructions tend to fail (e.g., nuanced tones like “sarcastic yet warm”)? Could examples be provided showing typical mismatches between intended emotion and synthesized delivery?
- Since the “conductor” can be replaced by any LLM, how stable are the results across different LLM families (e.g., Qwen vs Gemini vs GPT)? Does the performance scale linearly with the LLM’s instruction-following ability?
- Could the authors include more perceptual metrics (e.g., MOS, naturalness, style similarity) or statistical tests to assess whether improvements in emotion accuracy are perceptually significant?
- Could the authors provide examples showing how individual feature values map to specific expressive qualities (e.g., how changing pitch slope affects “sarcastic” tone)? This would strengthen the operationalist interpretation.

---

### Official Review · Reviewer_aFfi · 2025-11-01

**Soundness:** 3
**Presentation:** 3
**Contribution:** 2
**Rating:** 2
**Confidence:** 4

**Summary:**

The paper presents a workflow for emotional TTS that incorporates cues like pitch, energy, and timbre during inference. Specifically, these cues are from an external extractor during training and zero-shot LLM prediction during inference. Predicting speech features from LLM is the major contribution of this work.

Experimentally, the proposed system shows moderate WER and promising emotion accuracy. The authors also show the effectiveness of preference alignment, the impact of different LLMs, and cross-lingual capability.

**Strengths:**

(1) The system contains a complete training pipeline that includes pre-training, post-training, and preference alignment. The reported numbers are fairly competitive on the emotion side.
(2) It is good to demonstrate that the super-large LLM (e.g., Gemini) is able to provide accurate cues for fine-grained emotion control in a zero-shot manner, which should be rarely seen in previous literature.

**Weaknesses:**

(1) Feasibility: The proposed method heavily relies on the scale of the text LLM, which is far beyond the reasonable size of a TTS application. As in Fig.3,  even Qwen3-80B can only achieve 39.8% accuracy, and the 1.7B model text LLM is very poor. This observation significantly compromises the feasibility of the proposed method.
(2) Besides the text transcription, the only additional input information is the energy, pitch, and timbre, which are used to guide the model to generate emotional speech. However, I think the author should justify that these features are sufficient to infer the emotion.
(3) Fair comparison in experiments:
(3.1) Since you are using a very large text LLM and your overall system size is super large, I would recommend comparing some TTS systems more than 7B.
(3.2) Table 3 compares CosyVoice, why not CosyVoice2? As your work is based on CosyVoice2.
(3.3) Conventionally, the pitch, timbre, and energy could be predicted by the model itself by some specific structure (e.g., fastspeech 2 and its variants). Instead of predicting these features with LLMs, the author should also try this approach for comparison, to justify its effectiveness.
(3.4) For all the baseline models, I recommend the authors to describe how they test these models: do they accept the same input as the proposed model? Also, are these models specialized in emotion speech? if not, I would recommend the authors to also compare with those specialized models.
(3.5) The authors use Gemini in Table 1 for both model inference and evaluation, which is not convincing and has the risk of hacking the judge model. Instead, I would recommend the authors to also report an accuracy based on a classical emotional classifier.
(4) The preference optimization selects the pairs based on WER and speech rate, which is not relevant to emotion. However, the author claims this helps the emotion accuracy, which is not consistent.

**Questions:**

As in weakness.

---

### Official Review · Reviewer_HPrD · 2025-11-10

**Soundness:** 2
**Presentation:** 3
**Contribution:** 2
**Rating:** 4
**Confidence:** 4

**Summary:**

This paper proposes a decoupling paradigm that separates the semantic understanding of high-level instructions from the generation of low-level acoustic features. The framework utilizes an external, large language model to convert the user's natural language instructions into a structured, text-based set of acoustic features. Subsequently, a TTS model specifically designed for this task receives this textualized acoustic feature set along with the text to be synthesized and generates the final speech waveform.Experimental results demonstrate that the framework achieves a good performance in terms of emotional controllability, and exhibits generalization ability on zero-shot cross-lingual control tasks.

**Strengths:**

1. The data processing workflow proposed in this paper bypasses the challenge of large-scale, expensive, and inconsistent manual instruction annotation for controllable TTS tasks by automatically extracting acoustic features from existing speech data to create "instruction-feature" pairs.
2. In zero-shot performance on the Chinese sentiment benchmark, its performance outperforms the CosyVoice and Minimax models optimized for Chinese, even when post-trained using only English data.

**Weaknesses:**

1. The entire framework relies excessively on Gemini-2.5-pro, from instruction rewriting to evaluation and test set construction. Based on my past experience using Gemini for annotation, I found that its judgment of acoustic information often has significant biases. The authors did not conduct verification experiments to address these potential biases, and Gemini's reliability is precisely the cornerstone of the entire work.

2. In preference optimization, speech rate and WER are used to select preferred answers during rejection sampling, but I am not sure how strongly these two metrics are correlated with the claimed naturalness and expressiveness.

3. Due to the instability of emotion accuracy evaluations by Gemini and the bias introduced by this framework’s reliance on a strong text model, the experimental results are not very convincing. Although it shows a large advantage over Minimax on the test set, its performance in human evaluation is actually worse than Minimax, so the current results are questionable

4. Is this comparison fair? Compared with other models, my understanding is that additional conditions were provided during both training and inference to fine-tune the model and help it reason. If other models also had Gemini-assisted decomposition of complex instructions, what would their performance look like?

**Questions:**

See above

---

### Note · Authors · 2025-12-31

I have read and agree with the venue's withdrawal policy on behalf of myself and my co-authors.